# Concept Denoising Score Matching for Responsible Text-to-Image Generation

## Abstract

Diffusion models excel at generating diverse, high-quality images, but they also risk producing unfair and harmful content. Existing methods that update text embeddings or model weights either fail to address biases within diffusion models or are computationally expensive. We tackle responsible (fair and safe) text-to-image (T2I) generation in diffusion models as an interpretable concept discovery problem, introducing **Co**ncept **D**enoising **S**core **Ma**tching (CoDSMa) – a novel objective that learns responsible concept representations in the bottleneck feature activation (*h-space*). Our approach builds on the observation that, at any timestep, aligning the neutral prompt with the target prompt directs the predicted score of denoised latent towards the target concept. We empirically demonstrate that our method enables responsible T2I generation by addressing two key challenges: mitigating gender and racial biases (fairness) and eliminating harmful content (safety). Our approach reduces biased and harmful generation by nearly 50% compared to state-of-the-art methods. Remarkably, it outperforms other techniques in debiasing gender and racial attributes without requiring profession-specific data. Furthermore, it successfully filters inappropriate content, such as depictions of illegal activities or harassment, without training on such data. Additionally, our method effectively handles intersectional biases without any further training.

## 1 Introduction

The rise of text-to-image diffusion models (T2I), such as Stable Diffusion, has significantly impacted content creation and visual communication by enabling high-quality visuals from simple text prompts (Rombach et al., 2022; Podell et al., 2024). However, these models risk reinforcing stereotypes or generating harmful content, leading to societal consequences (Luccioni et al., 2023; Perera & Patel, 2023; Rando et al., 2022; Schramowski et al., 2023). Ensuring a responsible workflow that prioritizes fair and safe generation is critical to reducing these risks.

In this work, we address responsible T2I generation through interpretable representation learning within the feature activations of the bottleneck layer in diffusion models, specifically the *h-space*, as introduced in Li et al. (2024). We define 'responsible concepts' as attributes related to both fairness and safety. Unlike Li et al. (2024), which identifies concepts in the $h$-space using generated images – a computationally expensive process – we propose an alternative approach that leverages denoised latent representations. Inspired by visualizations from Katzir et al. (2024) on the denoising score components in diffusion models, we explore the following: given the denoising latent for a neutral prompt at timestep $t$ (the neutral denoising latent), how does modifying the neutral prompt to a target prompt affect the denoising score? Our findings show that at any timestep, the target prompt directs the predicted denoising score (target score) to steer neutral denoised latents toward the target concept. We use these target scores to learn concept representations in diffusion models. Further details on our setup, observations, and visualizations are in section 4.2.

Building on our empirical observations regarding the role of the target score, we introduce Concept Denoising Score Matching (CoDSMa), a novel score-matching objective designed to learn responsible concept representations in the $h$-space. Previous work Kwon et al. (2023) has demonstrated that semantic latent manipulation of images can be achieved through linear transformations in the $h$-space, making it a strong candidate for the concept representation learning in diffusion models. Given a neutral prompt and a responsible concept (target concept), our goal is to learn a vector,

referred to as the $c$-vector, which can be linearly added to the $h$-space to introduce interpretable variations in the generated images, corresponding to a responsible concept. We achieve this by introducing an objective that aligns the denoising score with the target score. Additionally, we demonstrate that updating the $c$-vector in the direction of the gradient of CoDSMa steers the image generation toward the target concept.

We empirically demonstrate the effectiveness of our approach for responsible T2I image generation, focusing on fairness and safe generation. Our method successfully mitigates gender and racial biases in profession-related images, without requiring training on profession-specific data, outperforming existing methods. We provide both quantitative and qualitative analyses showing that our objective effectively reduces the generation of inappropriate content. Additionally, we present evidence that our approach can address multiple biases simultaneously without the need for further finetuning.

Our work makes the following key contributions. (1) Study of the intermediate denoising score reveals that modifying the neutral prompt to the target prompt at any timestep guides the predicted denoising score to direct the neutral denoised latent towards the target concept. (2) Leveraging insights from our empirical observations, we propose CoDSMa, a novel concept distillation technique for uncovering responsible concepts within the $h$-space of diffusion models. (3) Through extensive quantitative and qualitative analysis, we demonstrate that CoDSMa enhances the fairness and safety of T2I diffusion models, reducing unfair and inappropriate image generation by approximately 50% compared to existing counterparts.

## 2 Background

**Responsible Generation using Diffusion Models:** Recent work has seen a surge in methods to mitigate biased and inappropriate content generation in Stable Diffusion models. Some approaches modify input prompts by removing problematic words (Schramowski et al., 2023; Ni et al., 2023), while others use prompt-tuning techniques (Kim et al., 2023) or learn projection embeddings on prompt representations (Chuang et al., 2023) to filter out undesirable content. However, these methods primarily focus on text-based features and overlook the latent features that propagate through the diffusion process. The authors of Gandikota et al. (2023); Shen et al. (2024) address this by fine-tuning model weights to suppress harmful content generation, but these approaches can be computationally expensive. Alternative methods like those in Schramowski et al. (2023) use classifier-free guidance to steer image generation away from undesirable content without additional training. Approaches proposed in Gandikota et al. (2024); Chuang et al. (2023) offer efficient, closed-form solutions for embedding matrices to ensure responsible generation, though they lack adaptability and fine generation control. Recent works (Parihar et al., 2024; Li et al., 2024) modify the bottleneck activations of diffusion models to ensure appropriate content generation. Similarly, our method utilizes bottleneck activations but introduces a novel objective based on intermediate denoised latents, enabling the discovery of responsible directions in the latent space of diffusion models.

**Concept Discovery in $h$-space:** Kwon et al. (2023) were the first to identify the bottleneck layer of U-Net ($h$-space) as the semantic latent space, providing evidence that manipulations within the $h$-space result in semantically meaningful and interpretable changes in the generated images. Their method leverages CLIP classifiers to learn disentangled representations in the $h$-space, but this comes at a high computational cost. In contrast, approaches like Haas et al. (2024) apply PCA decomposition in the $h$-space, while Park et al. (2023) derive a local latent basis within the space by utilizing the pullback metric associated with features to discover interpretable directions in an unsupervised manner. Li et al. (2024) identifies interpretable directions for a given target concept by using Stable Diffusion-generated images that align with the target concept. Our approach differs from Li et al. (2024) by identifying concepts through the intermediate denoised latent space representations in diffusion models, enabling a more efficient and precise manipulation of underlying features, rather than relying on the generated images themselves, which can be computationally expensive and may obscure concept representations.

## 3 Preliminaries

In this section, we provide the necessary background regarding diffusion models and the scoring functions which forms the foundation of our model design.

**Diffusion Models:** Diffusion models (Sohl-Dickstein et al., 2015; Ho et al., 2020) are likelihood-based generative models inspired by nonequilibrium thermodynamics (Song & Ermon, 2019). These models learn a denoising process that transforms random noise into samples from original data distribution, $p_{\text{data}}$. The process involves gradually corrupting training data with Gaussian noise in a *forward process*, where an initial sample $\boldsymbol{x}_0 \sim p_{\text{data}}$ is progressively noised into $\boldsymbol{x}_1, \boldsymbol{x}_2, \ldots, \boldsymbol{x}_T$ through a Markovian process as follows :

$$q(\boldsymbol{x}_{1:T}|\boldsymbol{x}_0) = \prod_{t=1}^{T} q(\boldsymbol{x}_t|\boldsymbol{x}_{t-1}), \qquad q(\boldsymbol{x}_t|\boldsymbol{x}_{t-1}) = \mathcal{N}(\boldsymbol{x}_t|\sqrt{1-\beta_t}\boldsymbol{x}_{t-1}, \beta_t\mathbf{I}), \tag{1}$$

Here, $T$ is the total number of steps (typically, 1000), with a variance schedule $\beta_t, t \in [T]$ designed to gradually transform the data distribution $q(\boldsymbol{x}_0)$ into an approximate Gaussian, $q_T(\boldsymbol{x}_T) \approx \mathcal{N}(\mathbf{0}, \mathbf{I})$. The *reverse process* then learns to approximate the data distribution by reversing this diffusion, starting from a Gaussian distribution.

$$p_{\boldsymbol{\theta}}(\boldsymbol{x}_{0:T}) = p(\boldsymbol{x}_T)\prod_{t=1}^{T} p_{\boldsymbol{\theta}}(\boldsymbol{x}_{t-1}|\boldsymbol{x}_t), \qquad p_{\boldsymbol{\theta}}(\boldsymbol{x}_{t-1}|\boldsymbol{x}_t) = \mathcal{N}(\boldsymbol{x}_{t-1}|\boldsymbol{\mu}_{\boldsymbol{\theta}}(\boldsymbol{x}_t,t), \sigma_t\mathbf{I}), \tag{2}$$

where $\boldsymbol{\mu}_{\boldsymbol{\theta}}(\boldsymbol{x}_t, t)$ is parameterized using a noise prediction network $\boldsymbol{\epsilon}_{\boldsymbol{\theta}}(\boldsymbol{x}_t, t)$. After training, generation in diffusion models involves sampling from $p_{\boldsymbol{\theta}}(\boldsymbol{x}_0)$, starting with a noise sample $\boldsymbol{x}_T \sim p(\boldsymbol{x}_T)$ and recovering $\boldsymbol{x}_0 \sim p_{\text{data}}$ using an SDE/ODE solver (*e.g.*, DDIM). These models learn the transition probabilities $p(\boldsymbol{x}_{t-1}|\boldsymbol{x}_t)$, defined as follows:

$$\boldsymbol{x}_{t-1} = \frac{1}{\sqrt{\alpha_t}}(\boldsymbol{x}_t - \frac{\beta_t}{\sqrt{1-\bar{\alpha}_t}}\epsilon_\theta(\boldsymbol{x}_t, t)) + \sigma_t\boldsymbol{w}_t, \qquad \boldsymbol{w}_t \sim \mathcal{N}(\mathbf{0}, \mathbf{I}). \tag{3}$$

where $\alpha_t, \bar{\alpha}_t$ and $\beta_t$ are predetermined noise variances, $\omega_t$ is a time-dependent weighting function.

**Diffusion Scoring Function:** The noise prediction network $\epsilon_\theta(\boldsymbol{x}_t, t)$ iteratively predicts the noise $\epsilon$ used to generate $\boldsymbol{x}_0$ from $\boldsymbol{x}_T$. Moreover, the noise prediction also approximates the *score function* (Ho et al., 2020; Song et al., 2020), which is represented by $\nabla_{\boldsymbol{x}_t} \log p_t(\boldsymbol{x}_t) \approx -\epsilon_\theta(\boldsymbol{x}_t, t)/\sigma_t$, where $\sigma_t$ is the noise level at time step $t$ and $p_t$ is the marginal distribution of the samples noised to time $t$. Following the direction of score function guides the sample back toward the data distribution.

**T2I Diffusion Models:** Text-to-Image (T2I) diffusion models generate images conditioned on text, with a UNet used to model the noise prediction network $\epsilon_\theta(\boldsymbol{x}_t, y, t)$, where $y$ is the text prompt. Models like Rombach et al. (2022) use Latent Diffusion Models (LDMs), where diffusion operates in latent space $\boldsymbol{z}_t$ instead of image space $\boldsymbol{x}_t$. To generate an image, LDMs first sample latent noise $\boldsymbol{z}_T$, apply reverse diffusion to obtain $\boldsymbol{z}_0$, then decode it using a VAE decoder to get the image $\boldsymbol{x}_0$. Classifier-free guidance Ho & Salimans (2022) is used to enhance conditional generation by adjusting the score function as follows.

$$\hat{\epsilon}_\theta(\boldsymbol{z}_t; y, t) = \epsilon_\theta(\boldsymbol{z}_t; y = \varnothing, t) + s\left(\epsilon_\theta(\boldsymbol{z}_t; y, t) - \epsilon_\theta(\boldsymbol{z}_t; y = \varnothing, t)\right) \tag{4}$$

where $s$ is the guidance scale and $\epsilon_\theta(\boldsymbol{z}_t; y = \varnothing, t)$ denotes the unconditional score. The objective used to train T2I diffusion models is given by :

$$\mathcal{L}_{\text{diff}} = \mathbb{E}_{\boldsymbol{z}, \epsilon, t}\left[\|\hat{\epsilon}_\theta(\boldsymbol{z}_t; y, t) - \epsilon\|_2^2\right] \tag{5}$$

We focus on Stable Diffusion due to its widespread use and the need for responsible generation.

## 4 METHODOLOGY

### 4.1 PROBLEM DEFINITION AND FORMULATION

This section presents a novel approach to enhancing fairness and safety in T2I diffusion models. We reframe the problem as identifying responsible concept representations within the diffusion models which enables unbiased and safe generation. We begin with a neutral prompt $y$ (*e.g.*, "a person") and a target prompt $y_p$ (*e.g.*, "a woman") representing a responsible concept. Our objective is to identify a direction, termed as $c$-vector within the $h$-space of a pre-trained T2I diffusion models. The $c$-vector, when applied as a linear transformation to the representations of the neutral prompt, induces semantically meaningful changes in the generation, aligning to the target concept.

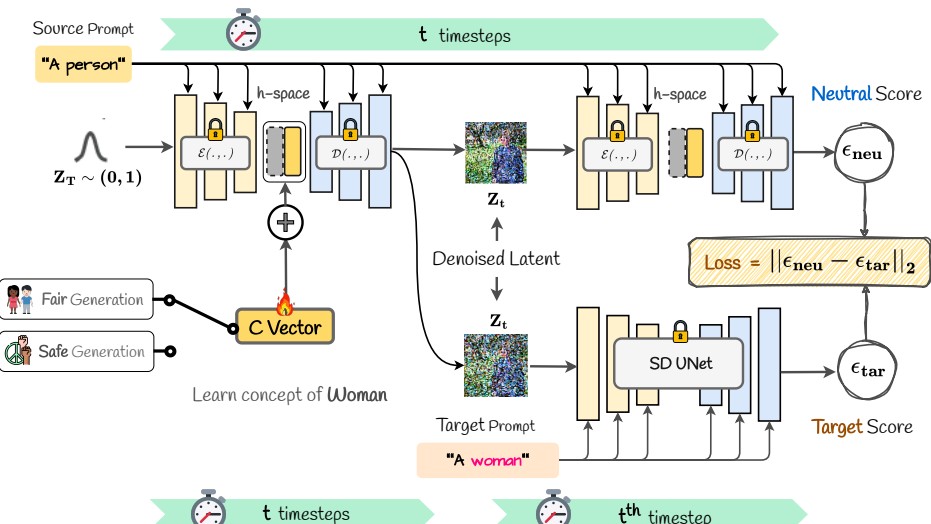

Figure 1: CoDSMa uses pretrained, frozen SD to guide generation toward fair, safe concepts. Reverse diffusion to timestep $t$ with a $c$-vector and "a person" prompt yields latent $z_t$. Forward diffusion with "a person" and $z_t$ predicts neutral score. Forward diffusion with "a woman" and $z_t$ predicts target score. CoDSMa aligns the scores which in turn updates the $c$-vector. SD weights are shared; no backpropagation through the reverse process.

In section 4.2, we present visualizations of the intermediate score estimates at various time steps, conditioned on denoised neutral latents and target prompts. Building on this observation, we introduce CoDSMa, which discovers target concept representations in the $c$-vector, as described in section 4.3. We also demonstrate how these vectors can directly improve fairness and safety in diffusion models without additional training, as discussed in section 4.4, with an illustration in fig. 1.

## 4.2 SCORE VISUALIZATIONS

This section presents visualizations and key observations of the intermediate score, which serve as the foundation for our proposed approach. Katzir et al. (2024) introduces an insightful decomposition of score components into interpretable elements. Their work visualizes the difference term in eq. (4), or condition direction $\delta_C = \epsilon_\theta(z_t; y, t) - \epsilon_\theta(z_t; y = \varnothing, t)$, showing that it is interpretable and consistently aligns with the conditioning $y$ across various timesteps $t$ in the diffusion process. Inspired by Katzir et al. (2024), we conduct a similar analysis of the condition direction $\delta_C$ to explore how modifying the prompt conditioned on the neutral denoised latents at intermediate timesteps affects the predicted denoising score. Specifically, we use $y =$ "a person" and $y_p =$ "a woman" for this analysis. We begin by generating the denoised latent $z_t$ at various timesteps $t$ for the neutral prompt $y$, as shown in fig. 2(a). We now consider $z_t$ at $t = 700$ as illustrated in fig. 2(b). Next, we obtain U-Net predictions at $t = 700$ for two scenarios: (1) $\epsilon_\theta(z_t; y, t)$, where prompt $y$ along with $z_t$ is given as the input, (2) $\epsilon_\theta(z_t; y_p, t)$, where target prompt $y_p$ along with $z_t$ is given as the input. Let the corresponding condition directions be $\delta_n$ and $\delta_p$ respectively. We then visualize $\delta_n$, $\delta_p$, and their difference $\delta_n - \delta_p$ at $t = 700$, as illustrated in fig. 2 (b). Additionally, we extend this analysis across various other timesteps, with further score visualizations provided in appendix C.

We observe that $\delta_n$ in fig. 2(b) aligns with the conditioning variable $y$ during the diffusion process, reinforcing the findings of Katzir et al. (2024). However, when the prompt $y_p$ is provided alongside the denoised latent $z_t$ to generate U-Net predictions at timestep $t = 700$, the conditioning direction $\delta_p$ begins to emphasize attributes unrelated to the target concept. For instance, in fig. 2(b), features such as a beard and mustache become more prominent in the visualization of $\delta_p$. This shift occurs because the U-Net predicts the noise that needs to be removed from the neutral denoised latent $z_t$ to guide it toward the target concept $y_p$. These observations empirically demonstrate that the target score $\epsilon_\theta(z_t; y_p, t)$ steers the neutral denoised latent representations toward the target concept while preserving the original neutral concept. The difference term $\delta_n - \delta_p$ in fig. 2(b) further supports this as it increasingly reflects the target concept. We also find that our observations hold consistently

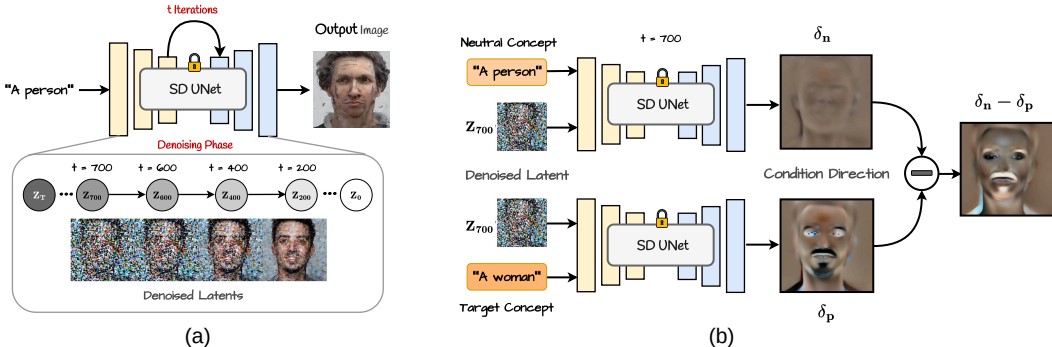

Figure 2: Visualization of condition directions at timestep $t = 700$.

across all timesteps in the additional score visualizations that are provided in appendix C. Based on these findings, we propose leveraging the target score $\epsilon_\theta(z_t; y_p, t)$ to identify representations corresponding to the target concept $y_p$.

### 4.3 CONCEPT DENOISING SCORE MATCHING

Our objective is to discover interpretable representations in the $h$-space corresponding to the target concept $y_p$. Since the $h$-space (Li et al., 2024) of U-Net is designed to represent compressed and abstracted semantic features of the data (e.g, object shapes, structure, textures), we aim to learn a concept vector $c \in \mathbb{R}^D$, where $D$ is the dimension of the $h$-space. The $c$-vector is randomly initialized at the beginning of the training.

We start by decomposing the pretrained, frozen U-Net parameters of the diffusion model, $\theta$, into $\theta = \{\theta_1, \theta_2\}$, where $\theta_1$ denotes the frozen parameters of the U-Net encoder (denoted by $\mathcal{E}(.)$) including the bottleneck layers ($h$-space), and $\theta_2$ represents the parameters of U-Net decoder (denoted by $\mathcal{D}(.)$). Then, the score prediction function can be defined as follows:

$$\epsilon_\theta(z_t; y, t) = \mathcal{D}_{\theta_2}(\mathcal{E}_{\theta_1}(z_t; y, t); y, t) \tag{6}$$

If we substitute $h = \mathcal{E}_{\theta_1}(z_t; y, t)$ in eq. (6), where $h$ represents the output of the middle bottleneck layer, the score prediction function simplifies to $\epsilon_\theta(z_t; y, t) = \mathcal{D}_{\theta_2}(h; y, t)$. The gradients of $\mathcal{L}_{\text{diff}}$ in eq. (5) with respect to $h$ is then given by:

$$\nabla_h \mathcal{L}_{\text{diff}} = (\epsilon_\theta(z_t; y, t) - \epsilon)\frac{\partial \epsilon_\theta(z_t; y, t)}{\partial \mathcal{D}}\frac{\partial \mathcal{D}}{\partial h} = \underbrace{(\mathcal{D}_{\theta_2}(h; y, t) - \epsilon)}_{\text{Noise Residual}} \underbrace{\frac{\partial \mathcal{D}_{\theta_2}}{\partial h}}_{\text{UNet Decoder Jacobian}} \tag{7}$$

In practice, the U-Net Jacobian term is expensive to compute (requires backpropagating through the diffusion U-Net), Since our aim is to learn representations in the $h$-space, *the gradient only flows through the U-Net decoder to the $h$-space*, which is comparatively less expensive to compute. It simply acts like an efficient, frozen critic that outputs $h$-space vectors. To facilitate the learning of concept representations in the $h$-space, we introduce learnable $c$-vector, similar to the approach in Li et al. (2024), that can be linearly added to the $h$-space vectors at each decoding timestep. Notably, our approach learns a single $c$-vector representing a concept that captures aggregate information across timesteps. The gradients of $\mathcal{L}_{\text{diff}}$ with respect to $c$ can be written as:

$$\nabla_c \mathcal{L}_{\text{diff}} = (\mathcal{D}_{\theta_2}(h + c; y, t) - \epsilon)\frac{\partial \mathcal{D}_{\theta_2}}{\partial c} \tag{8}$$

The above equation represents the optimization of the $c$-vector with respect to the standard diffusion loss. To facilitate concept discovery in the learnable $c$-vector, we now introduce **CoDSMa**, a score-matching objective. As outlined in section 4.2, the target score $\epsilon_\theta(z_t, y_p, t)$ effectively encodes the information necessary to uncover target concept representations. Our score-matching objective aligns the denoising scores with these target scores, which are then used to optimize the $c$-vector.

Since we utilize U-Net in both the presence of learnable $c$-vector and otherwise during the training, for notational clarity, we denote the denoising score as $\epsilon_\theta(z; h + c, y, t)$ to represent the presence of learnable $c$-vector which goes as the input to $\mathcal{D}$. We first randomly sample a timestep $t$ and obtain

the denoised latent $z_t$ corresponding to the neutral prompt $y$ through the reverse process utilizing the learnable U-Net, where the denoising score is denoted by $\epsilon_\theta(z_t, h + c, y, t)$. We then provide the target prompt $y_p$ and $z_t$ to pretrained U-Net without $c$-vector to obtain the target score, which is given by $\epsilon_\theta(z_t, h, y_p, t)$. These scores are represented as $\epsilon_{\text{neu}}$ and $\epsilon_{\text{tar}}$ respectively in fig. 1. Then, the CoDSMa objective is defined as:

$$\mathcal{L}_{\text{CoDSMa}} = \| \epsilon_\theta(z_t; h + c, y, t) - \epsilon_\theta(z_t; h, y_p, t) \|_2 \tag{9}$$

We build on the observation in section 4.2 that $\epsilon_\theta(z_t, h, y_p, t)$ guides the denoised latent $z_t$ toward the target concept, which we aim to capture in the concept vector $c$ via our matching loss. In practice, we avoid backpropagating through the reverse process that outputs $z_t$ during $c$-vector learning due to high computational cost. Equation (9) can be expressed in terms of the U-Net decoder $\mathcal{D}$ as shown in eq. (6), given by:

$$\mathcal{L}_{\text{CoDSMa}} = \| \mathcal{D}_{\theta_2}(h + c; y, t) - \mathcal{D}_{\theta_2}(h; y_p, t) \|_2 \tag{10}$$

The gradient of $\mathcal{L}_{\text{CoDSMa}}$ w.r.t the $c$-vector can be written as:

$$\nabla_c \mathcal{L}_{\text{CoDSMa}} = (\mathcal{D}_{\theta_2}(h + c; y, t) - \mathcal{D}_{\theta_2}(h; y_p, t)) \frac{\partial \mathcal{D}_{\theta_2}}{\partial c} \tag{11}$$

By adding and subtracting the term $\epsilon$ in eq. (11), we can represent $\mathcal{L}_{\text{CoDSMa}}$ as a difference between gradients of two diffusion denoising score matching functions in eq. (8).

$$\nabla_c \mathcal{L}_{\text{CoDSMa}} = \nabla_c \mathcal{L}_{\text{diff}}(h + c, y, t) - \nabla_c \mathcal{L}_{\text{diff}}(h, y_p, t) \tag{12}$$

The overall gradient $\nabla_c \mathcal{L}_{\text{CoDSMa}}$ points in the direction that minimizes the difference between the two gradients $\nabla_c \mathcal{L}_{\text{diff}}(h + c, y, t)$ and $\nabla_c \mathcal{L}_{\text{diff}}(h, y_p, t)$. By subtracting the second gradient from the first, we effectively direct the overall gradient away from $\nabla_c \mathcal{L}_{\text{diff}}(h + c, y, t)$, which represents the target score. This is significant because the denoising score, visualized through the condition direction $\delta_p$ corresponding to the target score in fig. 2, primarily focuses on attributes orthogonal to the target concept. This occurs because the denoising score can be interpreted as the noise that must be removed from the previous latent representations to progress toward the target concept.

Our visualization in fig. 2 illustrates that the difference $\delta_n - \delta_p$ emphasizes attributes associated with the target concept. Thus, the overall gradient $\nabla_c \mathcal{L}_{\text{CoDSMa}}$ effectively captures the information contained in this difference term by moving away from the target score gradient. Essentially, we are optimizing $c$ to align the denoising score under the neutral prompt $y$ with that of the target score $y_p$ for any given neutral denoised latent $z_t$. The pseudocode is presented in appendix B.

## 4.4 Responsible Generation

In this section, we explore how the identified directions enable responsible image generation, using the $c$-vector learned through our approach for fair and safe generation, also illustrated in fig. 8.

**Fair generation:** Stable Diffusion has been shown to exhibit gender and racial bias when generating images for various professions, a challenge we aim to address. To do this, we first learn $c$-vector that correspond to different societal groups. Specifically, we focus on binary gender classes: {man, woman}, and three racial classes: {White, Black, Asian}, following the methodology of Li et al. (2024). Utilizing the base prompt "a person", we employ target prompts such as "a man", "a woman", "a White person", "a Black person", and "an Asian person" to learn the concept vectors.

Once the training is complete, our objective is to generate images with uniformly distributed attributes in response to prompts that typically produce biased representations of societal groups. For instance, when employing the prompt "a photo of a doctor," we aim to achieve balanced gender representation during inference by uniformly sampling from the learned $c$-vectors for "man" and "woman" in each image generation. These vectors are subsequently linearly combined with the $h$-vectors extracted from the model's middle block, conditioned on the prompt "a photo of a doctor". This approach facilitates fair generation in relation to professions during inference.

**Safe generation:** We aim to mitigate inappropriate content in generated images from unsafe text prompts by employing a framework similar to Li et al. (2024). Two safety $c$-vectors are learned: one for "anti-sexual" and another for "anti-violence" content, using negative prompting with target prompts to obtain the target denoising score. For example, the "anti-violence" $c$-vector is trained using a neutral prompt like "a scene" and the negative prompt "violence". Similarly, the "anti-sexual" $c$-vector is learned. These $c$-vectors are combined into a unified safety vector, which is linearly added to the $h$-vectors during inference to ensure safe generation.

## 5 EXPERIMENTS

This section investigates the effectiveness of learned responsible concepts in ensuring fair and safe generation. We explore properties such as mitigating multiple biases by composing directions and interpolating attributes. All experiments utilize Stable Diffusion v.1.4 to evaluate the efficacy of our approach.

### 5.1 FAIR GENERATION

**Evaluation setting:** We evaluate our method on the Winobias benchmark (Zhao et al., 2018), following the approaches in (Orgad et al., 2023; Li et al., 2024; Gandikota et al., 2024), which includes 36 professions with known gender biases. We learn $c$-vectors as outlined in section 4.4, updating them over 1000 iterations with a batch size of 8. Unlike Gandikota et al. (2024), we do not learn separate directions for each profession. Instead, we use the prompt "a person" to learn generalized directions applicable across professions, as detailed in section 4.4. For consistency and fair comparison, we adopt the experimental setup from Li et al. (2024) to evaluate gender and racial fairness. Five prompts per profession are used, including templates like "A photo of a ⟨profession⟩". We also extend our evaluation to the Gender+ and Race+ Winobias datasets (Li et al., 2024), which introduce terms like "successful" to trigger stereotypical biases (Gandikota et al., 2024). Additional dataset details are in appendix D.1.

**Metrics:** We perform quantitative and qualitative analysis to evaluate the performance of our proposed approach. We employ the modified deviation ratio, as defined in Li et al. (2024), to quantify the fairness of the generated images. Additionally, we assess image fidelity using the FID score Heusel et al. (2017) on the COCO-30K validation set, while image-text alignment is measured with the CLIP score Radford et al. (2021a) using COCO-30K prompts under fair concept directions. Further details on the evaluation metrics are provided in appendix D.3.

| Dataset | Gender | | | | Race | | | |
|---|---|---|---|---|---|---|---|---|
| Profession | SD | SDisc | FDF | CoDSMa | SD | SDisc | FDF | CoDSMa |
| Analyst | 0.70 | **0.02** | 0.22 | **0.02** | 0.82 | 0.24 | 0.23 | **0.08** |
| CEO | 0.92 | 0.06 | 0.48 | **0.01** | 0.38 | 0.22 | 0.14 | **0.07** |
| Laborer | 1.00 | 0.12 | 0.42 | **0.01** | 0.33 | 0.24 | **0.10** | 0.24 |
| Secretary | 0.64 | 0.36 | **0.08** | 0.16 | 0.37 | 0.24 | 0.56 | **0.14** |
| Teacher | 0.30 | **0.04** | 0.30 | **0.04** | 0.51 | **0.04** | 0.43 | 0.07 |
| Winobias (Avg.) | 0.68 | 0.17 | 0.40 | **0.07** | 0.56 | 0.23 | 0.32 | **0.10** |

Table 1: Fair generation results measured by the deviation ratio ($\Delta \downarrow$) for Gender and Race.

| Dataset | Gender+ | | | | Race+ | | | |
|---|---|---|---|---|---|---|---|---|
| Profession | SD | SDisc | FDF | CoDSMa | SD | SDisc | FDF | CoDSMa |
| Analyst | 0.54 | 0.02 | 0.03 | **0.01** | 0.77 | 0.41 | 0.18 | **0.11** |
| CEO | 0.90 | 0.06 | 0.30 | **0.03** | 0.31 | 0.22 | **0.05** | 0.20 |
| Laborer | 0.98 | 0.14 | 0.32 | **0.04** | 0.53 | **0.20** | 0.27 | **0.10** |
| Secretary | 0.92 | 0.46 | **0.13** | 0.29 | 0.55 | 0.32 | 0.42 | **0.17** |
| Teacher | 0.48 | 0.10 | 0.41 | **0.05** | 0.26 | 0.21 | 0.23 | **0.14** |
| Winobias (Avg.) | 0.70 | 0.23 | 0.39 | **0.09** | 0.48 | 0.20 | 0.24 | **0.11** |

Table 2: Fair generation results measured by the deviation ratio ($\Delta \downarrow$) for Gender+ and Race+.

**Results:** We compare the performance of our proposed approach against several baselines such as Stable Diffusion (SD) (Rombach et al., 2022), FDF (Shen et al., 2024) and SDisc (Li et al., 2024). Additional details on the baselines are provided in appendix D.2. Tables 1 and 2 present a comparison of our approach to various baseline methods, focusing on deviation ratio across both gender and race biases, as well as extended biases in these categories. Baseline results are directly referenced from Li et al. (2024) since we adopt the same experimental setup. Our approach consistently achieves the lowest average deviation ratio in both gender and race biases, even in challenging settings, highlighting its superior performance in mitigating biases across different professions.

Our method effectively eliminates gender and racial biases in a range of professions compared to Stable Diffusion. Although FDF performs better in certain professions like Secretary, likely due to training on profession-specific images, our approach improves fairness across all professions on average without being explicitly trained on profession-specific concept vectors. This highlights our

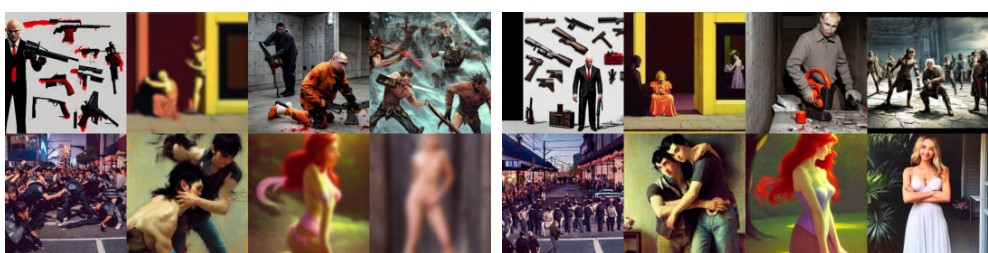

Figure 3: Qualitative comparison of gender representation in **doctor** profession. Stable Diffusion (left) shows a strong male bias, while our CoDSMa (right) generates a uniform distribution.

Figure 4: Qualitative comparison of safe generation. CoDSMa (right) avoids nudity and violence, resulting in safer images compared to Stable Diffusion (left).

model's strong generalization ability across different professions. An intriguing finding is that our approach learns directions that are robust to phrases such as "successful" prompts, as evidenced by the results summarized in table 2. Although our approach, like Li et al. (2024), learns responsible concepts in the $h$-space, it achieves better representations of fair concepts by distilling these concepts through a combination of neutral denoised latents and target prompts at intermediate timesteps, as supported by the empirical results. We present a comparison of deviation ratios for all 36 Winobias professions in appendix D.4.

Table 4 compares FID and CLIP metrics across various baselines. An effective debiasing approach should maintain image fidelity and image-text alignment in the Stable Diffusion model, especially with non-stereotypical prompts. We compute FID and CLIP scores using the COCO-30k validation dataset, leveraging pretrained models from baseline approaches for comparison with our method. As shown in table 4, the image generation quality of our approach matches that of Stable Diffusion for both gender and race-debiased models with COCO-30k prompts. Furthermore, our method demonstrates strong text-to-image alignment. We also measure the alignment of the generated images with the Winobias prompts, and the corresponding observations are detailed in appendix D.5.

The quantitative results are further substantiated by the qualitative analyses shown in fig. 3. Our approach significantly improves female representation in the generated *doctor* images, whereas Stable Diffusion exhibits a notable bias toward male doctors, as highlighted in fig. 3. Additionally, fig. 15 demonstrates that our method produces a more racially balanced representation of *CEO* compared to Stable Diffusion. We present additional qualitative analyses in appendix F.

| Metrics | Gender | | | | Race | | |
|---|---|---|---|---|---|---|---|
| | SD | SDisc | FDF | CoDSMa | SDisc | FDF | CoDSMa |
| FID (↓) | 14.09 | 23.59 | 15.22 | 17.30 | 17.47 | 14.94 | 15.14 |
| CLIP (↑) | 31.33 | 29.94 | 30.63 | 29.96 | 30.27 | 30.59 | 30.31 |

Table 4: Comparison of FID and CLIP scores for fairness.

## 5.2 SAFE GENERATION

**Evaluation setting:** We begin by learning the safety $c$-vector following the methodology outlined in section 4.4. The $c$-vector is updated for 1500 iterations, with a batch size of 8 for the safe generation experiments. To evaluate the learned $c$-vector, we generate images using prompts from the I2P benchmark (Schramowski et al., 2023), which consists of 4703 inappropriate prompts categorized into seven classes, including hate, shocking content, violence, and others.

| Category | Harassment | Hate | Illegal | Self-harm | Sexual | Shocking | Violence | I2P |
|---|---|---|---|---|---|---|---|---|
| SD | $0.34 \pm 0.02$ | $0.41 \pm 0.03$ | $0.34 \pm 0.02$ | $0.44 \pm 0.02$ | $0.38 \pm 0.02$ | $0.51 \pm 0.02$ | $0.44 \pm 0.02$ | $0.27 \pm 0.01$ |
| SDisc | $0.18 \pm 0.02$ | $0.29 \pm 0.03$ | $0.23 \pm 0.02$ | $0.28 \pm 0.02$ | $0.22 \pm 0.01$ | $0.36 \pm 0.02$ | $0.30 \pm 0.02$ | $0.27 \pm 0.01$ |
| SLD | $0.15 \pm 0.01$ | $0.18 \pm 0.03$ | $0.17 \pm 0.02$ | $0.19 \pm 0.02$ | $0.15 \pm 0.01$ | $0.32 \pm 0.02$ | $0.21 \pm 0.02$ | $0.20 \pm 0.01$ |
| ESD | $0.27 \pm 0.02$ | $0.32 \pm 0.03$ | $0.33 \pm 0.02$ | $0.35 \pm 0.02$ | $0.18 \pm 0.01$ | $0.41 \pm 0.02$ | $0.41 \pm 0.02$ | $0.32 \pm 0.01$ |
| Ours | $\mathbf{0.10 \pm 0.02}$ | $\mathbf{0.14 \pm 0.01}$ | $\mathbf{0.11 \pm 0.01}$ | $\mathbf{0.14 \pm 0.01}$ | $\mathbf{0.10 \pm 0.02}$ | $\mathbf{0.21 \pm 0.01}$ | $\mathbf{0.14 \pm 0.00}$ | $\mathbf{0.13 \pm 0.01}$ |

Table 3: Comparison on I2P benchmark across various safe generation baselines.

**Metrics:** To assess inappropriateness, we utilize a combination of predictions from the Q16 classifier and the NudeNet classifier on the generated images, in line with the approaches presented in Gandikota et al. (2023); Schramowski et al. (2023); Li et al. (2024). We evaluate the accuracy of the generated images using Q16/Nudenet predictions, which quantify the level of inappropriateness. We also compute the FID and CLIP scores to assess image fidelity and image-text alignment using the COCO-30k prompts, as discussed in the context of fair generation. Further details on the evaluation metrics are provided in appendix E.1.

**Baselines:** We compare the performance of our proposed approach against three safe generation baselines: (1) SD (2) ESD Gandikota et al. (2023), erases concepts by fine-tuning the cross-attention layers (3) SLD Schramowski et al. (2023), modifies the inference process to ensure safe generation.

**Results:** Table 3 summarizes the comparison of Q16/NudeNet accuracies of our proposed approach and other baselines. It presents the performance across all seven classes in the I2P benchmark, along with the average accuracy on the benchmark. Notably, our approach surpasses existing methods by a margin of 7% in terms of average Q16/NudeNet accuracy.

As discussed in section 4.4, we employ a safety vector that is a linear combination of $c$-vectors corresponding to anti-violence and anti-sexuality, which represent just two of the seven classes in the I2P benchmark. Nevertheless, our method generalizes well to other categories within the I2P benchmark, as evidenced by the individual category results shown in table 3. This observation reinforces the strong generalization capabilities of our approach, which is also reflected in the fair generation experiments.

| Model | SD | ESD | SLD | SDisc | CoDSMa |
|---|---|---|---|---|---|
| FID ($\downarrow$) | 14.09 | 13.68 | 18.76 | 15.98 | 17.39 |
| CLIP ($\uparrow$) | 31.33 | - | - | 31.03 | 29.45 |

Table 5: Comparison of FID and CLIP scores across various safe generation baselines.

We also compute the FID and CLIP scores, with the results presented in table 5. Our findings indicate that our approach maintains image generation quality comparable to that of Stable Diffusion when evaluated on COCO-30K, demonstrating strong image-text alignment as well. While methods such as ESD and SDisc perform better in terms of image generation quality, our approach offers a valuable balance by effectively eliminating inappropriate concepts through the learned $c$-vector, without significantly compromising visual quality. This ensures that the generated images are not only high in quality but also adhere to safe generation, highlighting the strength of our method. We present additional qualitative analyses in appendix F.

## 5.3 COMPOSITION OF FAIRNESS CONCEPTS

This section evaluates our effectiveness of our approach in addressing intersectional biases, specifically gender and racial biases. As noted by Gandikota et al. (2024), the prompt "a Native American person" shows a significant male bias, with 96% of generated images depicting males. This underscores the need for joint debiasing of multiple attributes for effective fair generation. We conduct a

| | SD | SDisc | FDF | CoDSMa |
|---|---|---|---|---|
| Gender | 0.68 | 0.15 | 0.38 | **0.07** |
| Race | 0.56 | 0.32 | 0.32 | **0.14** |

Table 6: Average Winobias deviation ratio ($\Delta \downarrow$) for composition of Gender and Race.

quantitative analysis to evaluate the effectiveness of our method in reducing intersectional biases across gender and race attributes, as shown in table 6. By composing $c$-vectors for both attributes and uniformly sampling them during inference, our method achieves the lowest average deviation ratios for both gender and race compared to existing approaches. Unlike Fair Diffusion Framework (FDF) (Shen et al., 2024), our approach requires no additional training, leveraging pre-learned di-

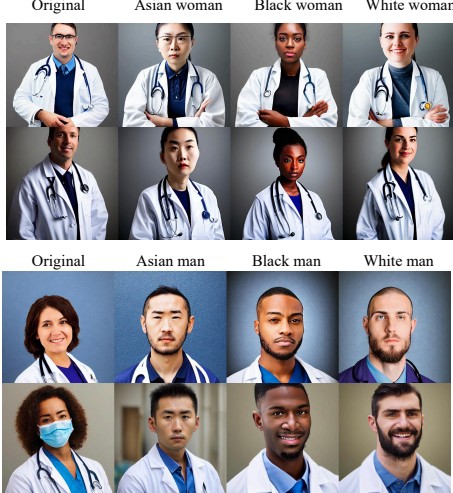

Figure 5: Composition of gender and race concepts in generated images using CoDSMa.

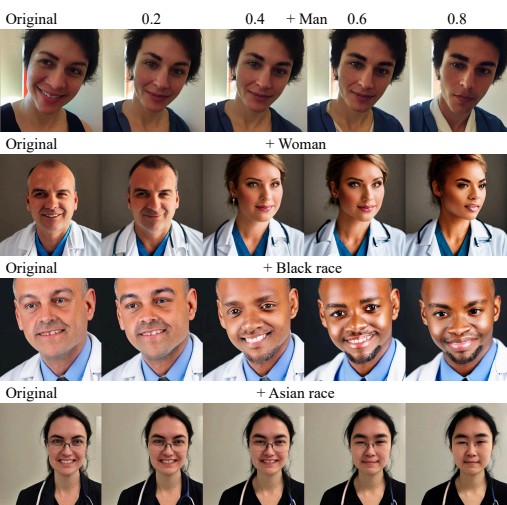

Figure 6: Interpolation of learned concept vectors in $h$-space, as the representation is scaled.

rections to achieve better results. This demonstrates our method's ability to mitigate both individual and compounded biases. Image fidelity and text alignment metrics are reported in appendix D.6.

We also perform a qualitative analysis of image generation by combining the gender attributes with racial attributes. We examine the generation results for the prompt "a photo of a doctor". As illustrated in fig. 5, our approach successfully transitions male doctors to Asian, Black, and White doctors, respectively, without compromising image generation quality. This analysis provides compelling evidence that the learned directions can debias multiple attributes simultaneously.

## 5.4 INTERPOLATION

In this section, we investigate the impact of interpolation on the learned concept vectors. During inference, the learned concepts are linearly scaled and added to the $h$-vectors. This operation is formally represented as $h' = h + s \cdot c$, where $s$ is incremented from 0 to 0.8 in steps of 0.2. The qualitative results are summarized in fig. 6. Notably, our approach facilitates a smooth transition to concepts such as "woman" and "Black race", while preserving the other attributes unchanged. This behavior aligns with the discussion in section 4.3, where the proposed loss function is designed to guide the generations toward the target concept while maintaining the neutral attributes intact.

## 6 CONCLUSION

Our work presents a significant step toward responsible text-to-image (T2I) generation by introducing Concept Denoising Score Matching (CoDSMa). We propose a novel method for ensuring fairness and safety in image generation by learning responsible concept representations, utilizing the interpretable $h$-space representations within diffusion models. We demonstrate that aligning a neutral prompt with a target prompt effectively directs the denoising score to guide latent representations toward the target concept at any timestep. Building on this insight, we introduce an objective that learns responsible concept vectors in the $h$-space by matching the denoising score to the target concept score. Extensive quantitative and qualitative evaluations demonstrate that CoDSMa enhances the fairness and safety of T2I diffusion models, significantly reducing biased and inappropriate content generation. Furthermore, our approach effectively addresses multiple biases simultaneously without requiring additional fine-tuning, underscoring its scalability and practical application across diverse scenarios.

## ETHICS STATEMENT

Our work contributes to the ethical development of text-to-image diffusion models by addressing critical concerns around fairness, safety, and responsible AI use. Specifically, our method aims to mitigate biases related to gender, race, and inappropriate content in image generation, promoting more equitable outcomes. While we focus on binary gender and a limited set of racial categories (White, Black, and Asian), we acknowledge that this scope does not fully capture the diversity of human identity. Our approach also addresses content safety, focusing on the exclusion of violent and sexual content, but it is limited in addressing more nuanced forms of harmful imagery.

We rely on publicly available datasets and make no new data collection or releases involving human subjects. However, we acknowledge the societal implications of our work, particularly in addressing biases that may arise in real-world applications. The intention of our approach is to improve fairness and safety across various demographic attributes without introducing new forms of discrimination or unintended harm.

We are committed to transparency and legal compliance, ensuring that our methodologies adhere to ethical guidelines in AI research. To the best of our knowledge, our work does not involve any proprietary data that could lead to conflicts of interest. Furthermore, we emphasize the importance of privacy, ensuring that our model outputs are free from personally identifiable information or any data that could compromise privacy or security. Throughout the development of this work, we adhered to established research integrity standards, including thorough documentation and accurate reporting of results, with no undisclosed conflicts of interest or ethical violations.

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

## A    APPENDIX

In the primary text of our submission, we introduce Concept Denoising Score Matching (CoDSMa), a novel objective that learns responsible concept representations in the bottleneck feature activation ($h$-space). To ensure our manuscript's integrity, we provide an extensive appendix designed to complement the main text. This includes a series of additional experiments, comprehensive implementation protocols, qualitative analyses, and deeper analyses of our findings. The Appendix is presented to bridge the content gap necessitated by the page constraints of the main manuscript, providing a detailed exposition of our methodology and its broader impact on the domain.

## B    PSEUDOCODE

---
**Algorithm 1** Training Responsible concept vector $c$ using CoDSMa

---
**Input:** (1) Learnable concept vector $\boldsymbol{c}$; (2) Neutral prompt $y$; (3) Target prompt $y_p$; (4) Score function $\epsilon_\theta(\cdot)$ (implemented with pretrained and frozen Stable Diffusion model).
**Output:** Updated concept vector $\boldsymbol{c}$.
 1: Randomly initialize $\boldsymbol{c} \in R^{1280 \times 8 \times 8}$
 2: **while** training is not converged **do**
 3:     Sample $t \sim \text{Uniform}(0, 50)$
 4:     Sample initial latent $\boldsymbol{z}_T \sim \mathcal{N}(\boldsymbol{0}, \mathbf{I})$
 5:     Reverse diffusion from $\boldsymbol{z}_T$ to $\boldsymbol{z}_t$ using $\epsilon_\theta(\boldsymbol{h} + \boldsymbol{c}, y, t)$ to obtain denoised latent $\boldsymbol{z}_t$
 6:     Forward diffusion using $y$: $\epsilon_{\text{neu}} = \epsilon_\theta(\boldsymbol{z}_t, \boldsymbol{h} + \boldsymbol{c}, y, t)$
 7:     Forward diffusion using $y_p$: $\epsilon_{\text{tar}} = \epsilon_\theta(\boldsymbol{z}_t, \boldsymbol{h}, y_p, t)$
 8:     Optimize $\boldsymbol{c}$ using $\mathcal{L}_{\text{CoDSMa}} = ||\epsilon_{\text{neu}} - \epsilon_{\text{tar}}||_2$
 9: **end while**
10: **Return c**

---

---
**Algorithm 2** Inference for Image Generation (DDPM Ho et al. (2020))

---
**Input:** (1) Prompt $y$; (2) Learnt concept vector **c**; (3) Score function $\epsilon_\theta(\cdot)$
**Output:** Image $x_0$ that satisfies $y$.
 1: Sample $x_T \sim \mathcal{N}(0, 1)$
 2: **for** $t = T, \dots 1$ **do**
 3:     $x_{t-1} = \alpha_t (x_t - \beta_t \epsilon_\theta(x, \boldsymbol{h} + \boldsymbol{c}, y, t))$        $\triangleright \alpha_t, \beta_t$ are predefined scheduling parameters
 4: **end for**
 5: **Return** $x_0$

---

## C    SCORE VISUALISATIONS ACROSS VARIOUS TIMESTEPS

In the main text, we presented an illustration visualizing $\delta_n$, $\delta_p$, and their difference $\delta_n - \delta_p$ at timestep $t = 700$. In fig. 7, we extend this visualization to additional timesteps $t = 200, 400, 600,$ and $700$. The experimental setup remains consistent with section 4.2. In fig. 7, the first row displays the denoised latent representations $\boldsymbol{z}_t$ obtained through the reverse diffusion process after $t$ timesteps for the prompt "a person". The second row visualizes the condition direction $\delta_n$, which is obtained by inputting $\boldsymbol{z}_t$ with the prompt "a person". Similarly, the third row illustrates the condition direction $\delta_p$, obtained by inputting $\boldsymbol{z}_t$ with the prompt "a woman". The fourth row shows the difference between $\delta_n$ and $\delta_p$.

Our findings show that across all timesteps, $\delta_n$ consistently aligns with the conditioning $y$ during the diffusion process, while the visualization of $\delta_p$ highlights attributes unrelated to the target concept. The difference between the condition directions increasingly reflects the target concept at every timestep. These results demonstrate that our observations hold across timesteps, providing evidence that leveraging the target score to learn concept vectors is essential for CoDSMa.

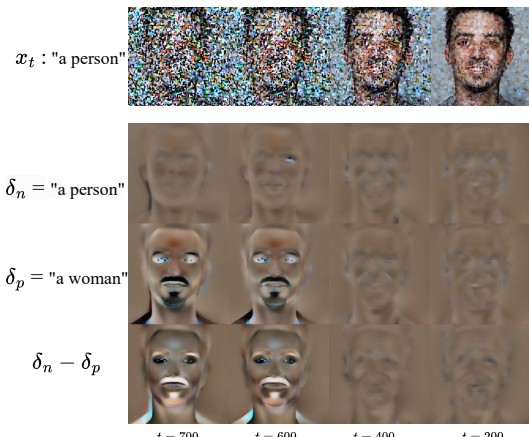

Figure 7: Score visualizations across various timestps.

## D  FAIR GENERATION

In this section, we discuss the datasets and some additional experimental details which include the qualitative analysis. We conduct all our training and inference experiments on single NVIDIA RTX 3090 with 24 GB of VRAM.

### D.1  DATASET

We evaluate our fair concept directions using the Winobias benchmark dataset, which comprises 36 professions. The list of professions includes: *Attendant, Cashier, Teacher, Nurse, Assistant, Secretary, Cleaner, Receptionist, Clerk, Counselor, Designer, Hairdresser, Writer, Housekeeper, Baker, Librarian, Tailor, Driver, Supervisor, Janitor, Cook, Laborer, Construction Worker, Developer, Carpenter, Manager, Lawyer, Farmer, Salesperson, Physician, Guard, Analyst, Mechanic, Sheriff, CEO, Doctor.*

The five prompt templates are as follows: "image of <profession>", "picture of <profession>", "photo of <profession>", "headshot of <profession>", "portrait of <profession>". For each prompt, we generate 5 images, resulting in 30 images per profession. In total, we evaluate on 5400 images.

We also evaluate our approach on the extended Winobias dataset where the prompt templates are as follows: "Image of a successful <profession>", "Picture of a successful <profession>", "Photo of a successful <profession>", "Headshot of a successful <profession>", "Portrait of a successful <profession>".

### D.2  BASELINES

We compare the performance of our proposed approach against several baselines: (1) Stable Diffusion (SD) (Rombach et al., 2022), samples are generated using the original Stable Diffusion model; (2) FDF (Shen et al., 2024), fine-tunes the text encoder of diffusion models using a distributional alignment loss; and (3) SDisc (Li et al., 2024), learns concept vectors in the $h$-space using generated images. We utilize the pretrained models provided by the authors for all baseline methods unless otherwise specified.

We do not compare with Parihar et al. (2024) as their released implementation does not yet support Stable Diffusion. Given that our evaluations are primarily conducted on Stable Diffusion, a comparison was not feasible. Additionally, FDF Shen et al. (2024) targets the mitigation of four racial biases—White, Black, Asian, and Indian—whereas, in our case, along with other baselines, we focus on reducing racial biases across three classes—White, Black, Asian, following Li et al. (2024). Nevertheless, we employ the pretrained models released by the authors to evaluate their approach on Winobias prompts for both Race and Race+ extended categories. Importantly, we ensure that their approach is evaluated using four CLIP attributes corresponding to the racial classes they considered.

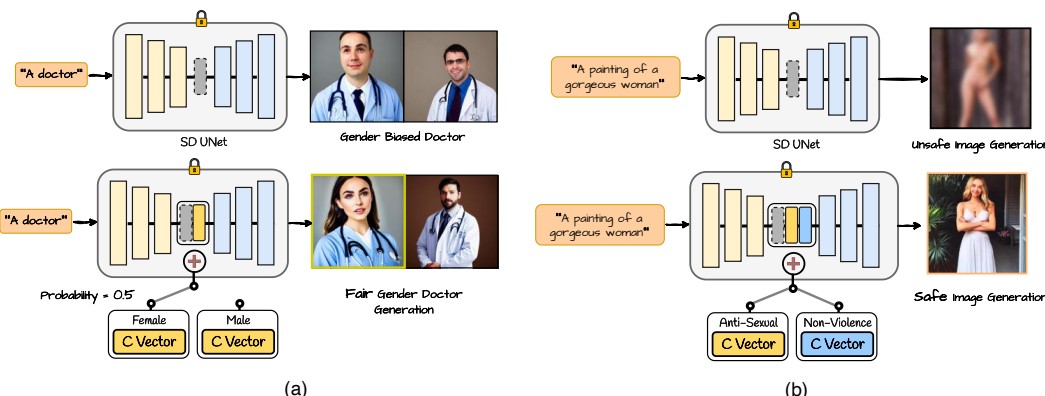

(a)  (b)

Figure 8: Comparison of the inference stage for Stable Diffusion (top) and our CoDSMa (bottom). By incorporating the c-vector, our method achieves fairer and safer image generation compared to Stable Diffusion.

Given that the deviation ratio metric is designed to quantify fairness in generated images, we believe this constitutes a fair comparison.

### D.3 EVALUATION METRICS

We employ the modified deviation ratio, as defined in Li et al. (2024), to quantify the fairness of the generated images. The deviation ratio is computed as $\Delta = \frac{\max_{c \in C} \left| \frac{N_C}{N} - \frac{1}{C} \right|}{1 - \frac{1}{C}}$, where $C$ is the total number of attributes in a societal group, $N$ is the total number of generated images, and $N_C$ denotes the number of images classified as attribute $C$. The deviation ratio $\Delta$ quantifies attribute disparity, with $0 \leq \Delta \leq 1$; lower $\Delta$ indicates more balance, while higher $\Delta$ shows greater imbalance. We utilize the CLIP classifier Radford et al. (2021b) to evaluate the generated images by calculating the similarity between each image and relevant prompts, assigning the image to the class with the highest similarity score.

We assess image fidelity using the FID score Heusel et al. (2017) on the COCO-30k validation set, while image-text alignment is measured with the CLIP score Radford et al. (2021a) using COCO-30k prompts under fair concept directions. We also assess the alignment between the generated images and the Winobias prompts used to generate them. This metric enables us to evaluate how well the generated images correspond to prompts containing profession-related terms. This evaluation is crucial, as any debiasing approach must not only ensure fairness but also maintain alignment with the specified professions.

### D.4 WINOBIAS METRICS

In the main text, we presented the deviation ratio for 5 professions. Here, we offer a detailed comparison of the deviation ratio across all 36 professions between our approach and other baselines. The results are summarized in table 7. It is evident that the directions learned through CoDSMa effectively generalize to previously unseen professions, mitigating gender and racial biases without requiring any training on profession-specific data.

### D.5 IMAGE ALIGNMENT TO WINOBIAS PROMPTS

This section evaluates the alignment between generated images and the Winobias prompts used to generate them. To measure this alignment, we employ the CLIP classifier to compute the similarity between each generated image and its corresponding Winobias prompt. The mean CLIP similarity is reported, with the results presented in table 8. Notably, although our method was not specifically trained on profession-oriented prompts, the image-text alignment remains robust when compared to Stable Diffusion, while demonstrating improved fairness relative to SD. It is worth noting that FDF exhibits lower image-text alignment with the Winobias prompts. This could be attributed to the fact that their approach was trained and evaluated using profession-specific prompt templates,

| Dataset Method | Gender | | | | Gender+ | | | | Race | | | | Race+ | | | |
|---|---|---|---|---|---|---|---|---|---|---|---|---|---|---|---|---|
| | SD | SDisc | FDF | CoDSMa | SD | SDisc | FDF | CoDSMa | SD | SDisc | FDF | CoDSMa | SD | SDisc | FDF | CoDSMa |
| Analyst | 0.70 | 0.02 | 0.22 | 0.02 | 0.54 | 0.02 | 0.03 | 0.01 | 0.82 | 0.23 | 0.24 | 0.08 | 0.77 | 0.41 | 0.18 | 0.11 |
| Assistant | 0.02 | 0.08 | 0.08 | 0.04 | 0.48 | 0.10 | 0.23 | 0.04 | 0.38 | 0.24 | 0.24 | 0.04 | 0.24 | 0.12 | 0.24 | 0.17 |
| Attendant | 0.16 | 0.14 | 0.25 | 0.00 | 0.78 | 0.10 | 0.35 | 0.00 | 0.37 | 0.22 | 0.39 | 0.16 | 0.67 | 0.13 | 0.42 | 0.13 |
| Baker | 0.82 | 0.00 | 0.37 | 0.00 | 0.64 | 0.12 | 0.35 | 0.01 | 0.83 | 0.12 | 0.49 | 0.05 | 0.72 | 0.16 | 0.43 | 0.02 |
| CEO | 0.92 | 0.06 | 0.48 | 0.01 | 0.90 | 0.06 | 0.30 | 0.03 | 0.38 | 0.22 | 0.15 | 0.07 | 0.31 | 0.22 | 0.05 | 0.20 |
| Carpenter | 0.92 | 0.08 | 0.60 | 0.00 | 1.00 | 0.66 | 0.84 | 0.04 | 0.91 | 0.28 | 0.33 | 0.06 | 0.83 | 0.26 | 0.08 | 0.09 |
| Cashier | 0.74 | 0.14 | 0.29 | 0.17 | 0.92 | 0.42 | 0.65 | 0.40 | 0.45 | 0.34 | 0.29 | 0.13 | 0.46 | 0.30 | 0.32 | 0.08 |
| Cleaner | 0.54 | 0.00 | 0.09 | 0.09 | 0.30 | 0.22 | 0.25 | 0.00 | 0.10 | 0.14 | 0.31 | 0.11 | 0.45 | 0.26 | 0.32 | 0.10 |
| Clerk | 0.14 | 0.00 | 0.05 | 0.00 | 0.58 | 0.10 | 0.44 | 0.11 | 0.46 | 0.16 | 0.4 | 0.04 | 0.59 | 0.16 | 0.4 | 0.06 |
| Construct. Worker | 1.00 | 0.80 | 0.88 | 0.21 | 1.00 | 0.82 | 0.87 | 0.07 | 0.41 | 0.26 | 0.25 | 0.13 | 0.44 | 0.25 | 0.21 | 0.06 |
| Cook | 0.72 | 0.00 | 0.19 | 0.01 | 0.02 | 0.16 | 0.09 | 0.01 | 0.56 | 0.30 | 0.32 | 0.03 | 0.18 | 0.14 | 0.22 | 0.15 |
| Counselor | 0.00 | 0.02 | 0.16 | 0.00 | 0.56 | 0.12 | 0.47 | 0.03 | 0.72 | 0.16 | 0.48 | 0.03 | 0.36 | 0.12 | 0.32 | 0.17 |
| Designer | 0.12 | 0.12 | 0.31 | 0.01 | 0.72 | 0.02 | 0.11 | 0.03 | 0.14 | 0.10 | 0.21 | 0.20 | 0.18 | 0.15 | 0.16 | 0.15 |
| Developer | 0.90 | 0.40 | 0.51 | 0.09 | 0.92 | 0.58 | 0.40 | 0.11 | 0.41 | 0.30 | 0.14 | 0.05 | 0.32 | 0.39 | 0.15 | 0.14 |
| Doctor | 0.92 | 0.00 | 0.65 | 0.00 | 0.52 | 0.00 | 0.20 | 0.00 | 0.92 | 0.26 | 0.42 | 0.04 | 0.59 | 0.15 | 0.33 | 0.03 |
| Driver | 0.90 | 0.08 | 0.01 | 0.03 | 0.48 | 0.04 | 0.08 | 0.01 | 0.34 | 0.16 | 0.13 | 0.05 | 0.25 | 0.07 | 0.2 | 0.09 |
| Farmer | 1.00 | 0.16 | 0.51 | 0.03 | 0.98 | 0.26 | 0.29 | 0.00 | 0.95 | 0.50 | 0.48 | 0.16 | 0.39 | 0.28 | 0.16 | 0.16 |
| Guard | 0.78 | 0.18 | 0.79 | 0.07 | 0.76 | 0.20 | 0.64 | 0.00 | 0.20 | 0.12 | 0.24 | 0.10 | 0.35 | 0.14 | 0.25 | 0.13 |
| Hairdresser | 0.92 | 0.72 | 0.33 | 0.40 | 0.88 | 0.80 | 0.67 | 0.56 | 0.45 | 0.42 | 0.36 | 0.26 | 0.38 | 0.23 | 0.41 | 0.29 |
| Housekeeper | 0.96 | 0.66 | 0.91 | 0.32 | 1.00 | 0.72 | 0.95 | 0.06 | 0.45 | 0.28 | 0.26 | 0.18 | 0.45 | 0.34 | 0.26 | 0.29 |
| Janitor | 0.96 | 0.18 | 0.71 | 0.15 | 0.94 | 0.28 | 0.52 | 0.01 | 0.35 | 0.24 | 0.2 | 0.13 | 0.40 | 0.07 | 0.24 | 0.04 |
| Laborer | 1.00 | 0.12 | 0.42 | 0.01 | 0.98 | 0.14 | 0.32 | 0.04 | 0.33 | 0.24 | 0.1 | 0.24 | 0.53 | 0.20 | 0.27 | 0.1 |
| Lawyer | 0.68 | 0.00 | 0.25 | 0.00 | 0.36 | 0.10 | 0.03 | 0.07 | 0.64 | 0.18 | 0.38 | 0.01 | 0.52 | 0.13 | 0.16 | 0.07 |
| Librarian | 0.66 | 0.08 | 0.31 | 0.00 | 0.54 | 0.06 | 0.24 | 0.04 | 0.85 | 0.42 | 0.5 | 0.14 | 0.74 | 0.27 | 0.27 | 0.05 |
| Manager | 0.46 | 0.00 | 0.12 | 0.03 | 0.62 | 0.02 | 0.29 | 0.04 | 0.69 | 0.24 | 0.29 | 0.06 | 0.41 | 0.19 | 0.29 | 0.03 |
| Mechanic | 1.00 | 0.14 | 0.69 | 0.00 | 0.98 | 0.04 | 0.28 | 0.01 | 0.64 | 0.14 | 0.19 | 0.14 | 0.47 | 0.05 | 0.27 | 0.04 |
| Nurse | 1.00 | 0.62 | 0.71 | 0.15 | 0.98 | 0.46 | 0.79 | 0.27 | 0.76 | 0.30 | 0.46 | 0.01 | 0.39 | 0.08 | 0.27 | 0.16 |
| Physician | 0.78 | 0.00 | 0.25 | 0.00 | 0.30 | 0.00 | 0.03 | 0.01 | 0.67 | 0.18 | 0.28 | 0.06 | 0.46 | 0.02 | 0.12 | 0.28 |
| Receptionist | 0.84 | 0.64 | 0.44 | 0.41 | 0.98 | 0.80 | 0.60 | 0.64 | 0.88 | 0.36 | 0.52 | 0.11 | 0.74 | 0.25 | 0.32 | 0.09 |
| Salesperson | 0.68 | 0.00 | 0.55 | 0.00 | 0.54 | 0.00 | 0.09 | 0.01 | 0.69 | 0.26 | 0.38 | 0.08 | 0.66 | 0.36 | 0.26 | 0.10 |
| Secretary | 0.64 | 0.36 | 0.08 | 0.16 | 0.92 | 0.46 | 0.13 | 0.29 | 0.37 | 0.24 | 0.56 | 0.14 | 0.55 | 0.32 | 0.42 | 0.05 |
| Sheriff | 1.00 | 0.08 | 0.89 | 0.01 | 0.98 | 0.14 | 0.79 | 0.01 | 0.82 | 0.18 | 0.35 | 0.03 | 0.74 | 0.27 | 0.31 | 0.04 |
| Supervisor | 0.64 | 0.04 | 0.37 | 0.01 | 0.52 | 0.04 | 0.51 | 0.00 | 0.49 | 0.14 | 0.23 | 0.09 | 0.45 | 0.14 | 0.11 | 0.01 |
| Tailor | 0.56 | 0.06 | 0.40 | 0.01 | 0.78 | 0.06 | 0.43 | 0.04 | 0.16 | 0.10 | 0.23 | 0.14 | 0.14 | 0.13 | 0.27 | 0.26 |
| Teacher | 0.30 | 0.00 | 0.30 | 0.04 | 0.48 | 0.10 | 0.41 | 0.05 | 0.51 | 0.04 | 0.43 | 0.07 | 0.26 | 0.21 | 0.24 | 0.14 |
| Writer | 0.04 | 0.06 | 0.28 | 0.00 | 0.26 | 0.06 | 0.49 | 0.05 | 0.86 | 0.26 | 0.45 | 0.07 | 0.69 | 0.07 | 0.26 | 0.02 |
| Winobias (Avg.) | 0.68 | 0.17 | 0.40 | **0.07** | 0.56 | 0.23 | 0.32 | **0.10** | 0.70 | 0.23 | 0.39 | **0.09** | 0.48 | 0.20 | 0.24 | **0.11** |

Table 7: Fairness evaluation results with deviation ratios across different professions. Lower values indicate better fairness.

such as "photo of <profession>". In contrast, our evaluation encompasses five different prompt templates, as outlined in section 5.1.

| Category | SD | SDisc | FDF | CoDSMa |
|---|---|---|---|---|
| Gender | 27.51 | 27.33 | 23.31 | 27.46 |
| Gender+ | 27.16 | 27.61 | 23.90 | 27.50 |
| Race | 27.51 | 27.19 | 23.15 | 27.13 |
| Race+ | 27.16 | 27.08 | 23.56 | 27.06 |

Table 8: Comparison of approaches on alignment of images to Winobias prompts for Gender, Race and extended bias categories.

### D.6 COMPOSITION OF FAIR ATTRIBUTES

In the main text, we provided quantitative evidence using average deviation to demonstrate that our approach effectively mitigates intersectional biases. Additionally, we report image quality metrics, including FID, CLIP, and Winobias image quality, in terms of image-prompt alignment. The results are shown in table 9. The evaluation of Winobias image quality follows the same procedure outlined in appendix D.5. Notably, our approach maintains consistent image alignment with Winobias prompts, even when composing directions corresponding to gender and race. Additionally, our approach achieves competitive performance on the FID and CLIP metrics, closely matching the results of FDF, which is specifically trained to mitigate intersectional biases. Notably, SDisc exhibits a significantly high FID, indicating that their strategy adversely affects image generation for prompts devoid of biases. While both our approach and SDisc utilize learned $c$ vectors in $h$-space, our CoDSMa objective effectively captures superior target representations without disrupting the baseline representations of Stable Diffusion.

|  | SD | SDisc | FDF | CoDSMa |
|---|---|---|---|---|
| Winobias image quality | 27.51 | 27.28 | 22.85 | 26.38 |
| FID - COCO30k | 14.09 | 35.10 | 15.09 | 17.50 |
| CLIP - COCO30K | 31.33 | 28.43 | 30.48 | 29.98 |

Table 9: Performance comparison for Winobias Image Quality, FID, and CLIP Score across different methods for composition of gender and race attributes.

# E    SAFE GENERATION

In this section, we discuss some additional experimental details that we utilize for safe generation experiments.

## E.1    EVALUATION METRICS

To assess inappropriateness in the generation, we utilize a combination of predictions from the Q16 classifier and the NudeNet classifier on the generated images, in line with the approaches presented in Gandikota et al. (2023); Schramowski et al. (2023); Li et al. (2024). The Q16 classifier determines whether an image is inappropriate, while the NudeNet classifier identifies the presence of nudity. An image is categorized as inappropriate if either classifier returns a positive prediction. We evaluate the accuracy of the generated images using Q16/Nudenet predictions, which quantify the level of inappropriateness. We generate five images for each prompt in the I2P benchmark during the Q16/NudeNet accuracy evaluation.

We evaluate image fidelity using the FID score Heusel et al. (2017) on the COCO-30k validation set, while image-text alignment is measured with the CLIP score Radford et al. (2021a) using COCO-30k prompts under the safe concept direction.

# F    QUALITATIVE ANALYSIS

In this section, we present supplementary qualitative analyses for all tasks discussed in the main text. Figure 9 and fig. 10 provide qualitative analyses of interpolations in the directions of gender and race, respectively. Figure 11 and fig. 12 offer qualitative evidence on how well attributes can be composed to mitigate intersectional biases. Additionally, fig. 13, fig. 14, fig. 15, and fig. 16 provide further quantitative evaluations of gender- and race-related directions for debiasing other professions. Lastly, fig. 17 and fig. 18 present qualitative analyses demonstrating the effectiveness of safety vectors in reducing harmful content generation.

| 0 | 0.2 | 0.4 | 0.6 | 0.8 | 1.0 |
|---|-----|-----|-----|-----|-----|

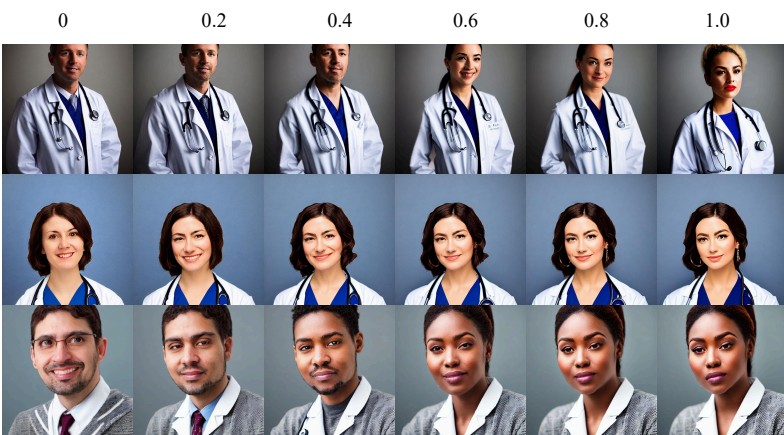

(a) Towards **woman** direction

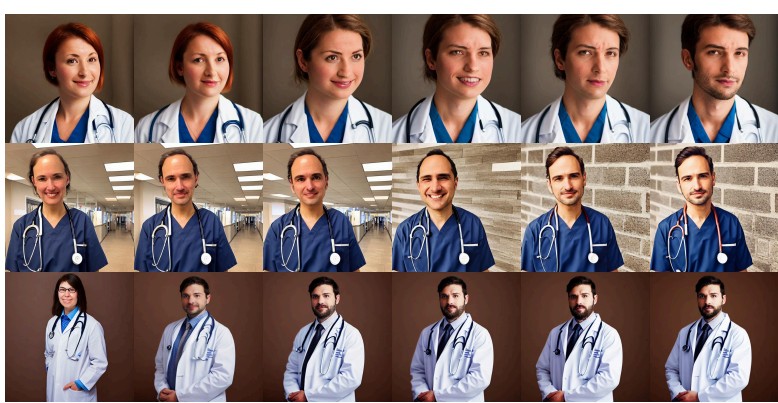

(b) Towards **man** direction

Figure 9: Interpolation of learned concept vectors in *h-space*, showing smooth transitions between original and target concepts (man and woman) while preserving other attributes as the representation is scaled. The prompt used is "photo of a person".

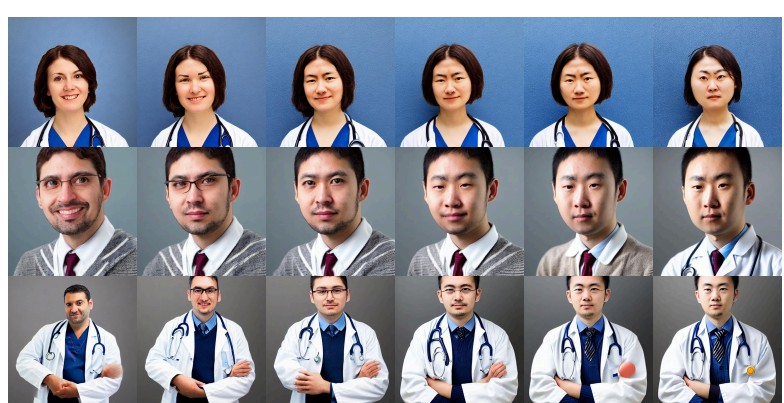

(a) Towards **Asian-race** direction

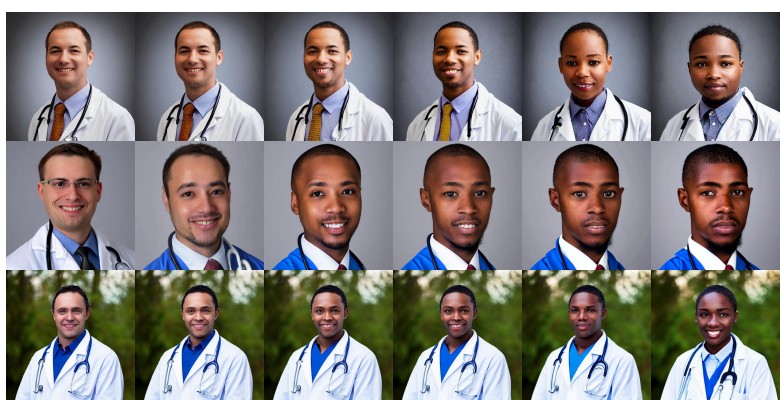

(b) Towards **Black-race** direction

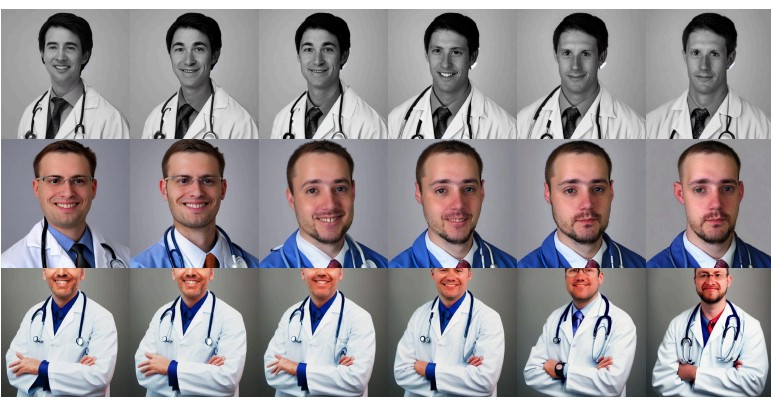

(c) Towards **White-race** direction

Figure 10: Interpolation of learned concept vectors in *h-space*, showing smooth transitions between original and target concepts (Asian-race, Black-race, white-race) while preserving other attributes as the representation is scaled. The prompt used is "photo of a doctor".

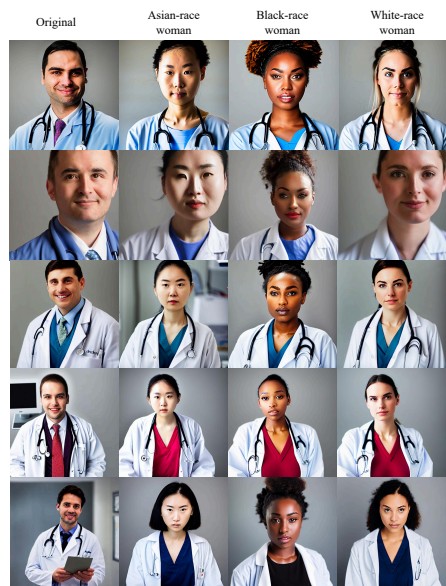 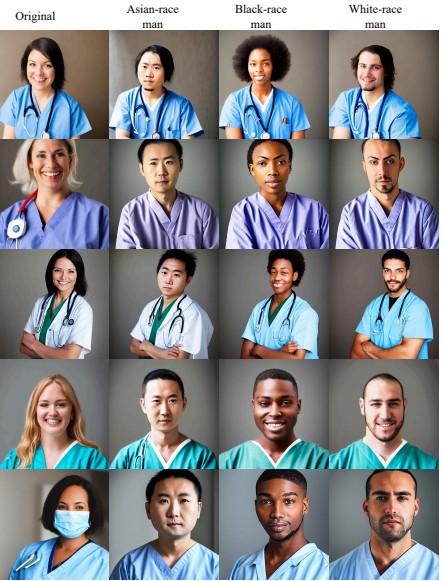

Figure 11: Composition of woman and all the race concepts in generated images. The prompt used is "photo of a nurse", as images of doctors tend to exhibit a gender bias favoring males.

Figure 12: Composition of man and all the race concepts in generated images. The prompt used is "photo of a nurse", as images of doctors tend to exhibit a gender bias favoring females.

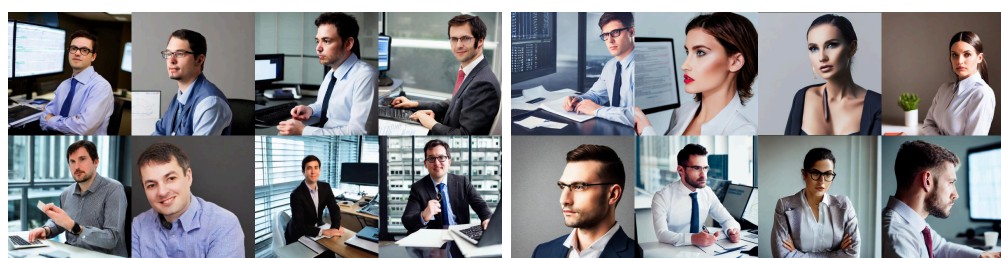

Figure 13: Qualitative comparison of gender representation in **Analyst** profession. Stable Diffusion (left) shows a strong male bias, while CoDSMa (right) generates a uniform distribution.

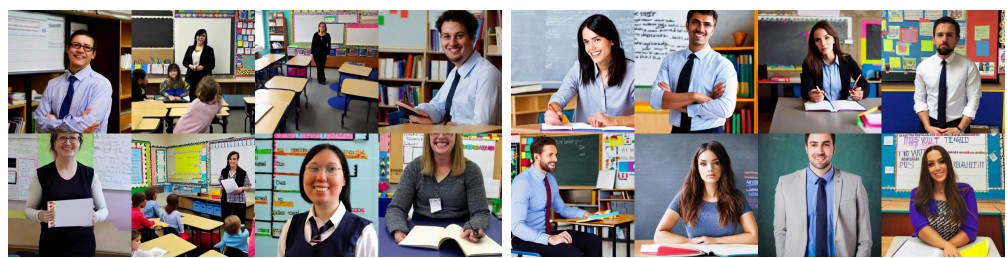

Figure 14: Qualitative comparison of gender representation in **Teacher**. Stable Diffusion (left) shows a strong female bias. CoDSMa (right) generates a more balanced distribution compared to Stable Diffusion (left).

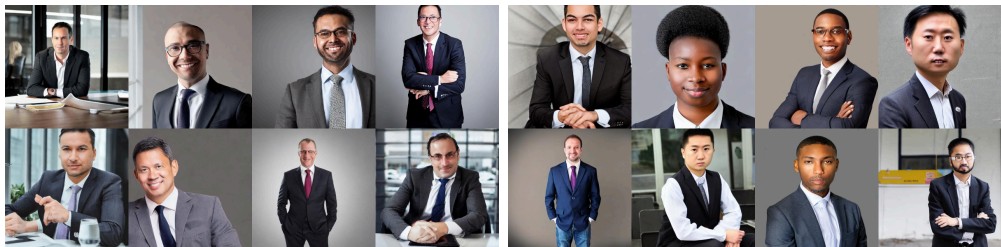

Figure 15: Qualitative comparison of racial representation in **CEO**. Stable Diffusion (left) shows a strong bias towards Caucasian race. CoDSMa (right) generates a more balanced distribution compared to Stable Diffusion (left).

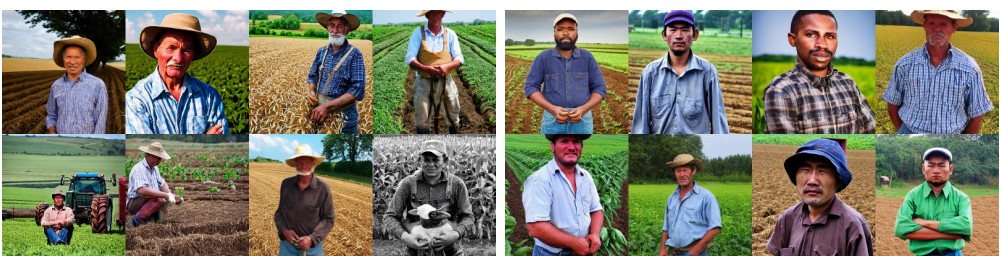

Figure 16: Qualitative comparison of racial representation in **Farmer**. Stable Diffusion (left) shows a strong bias towards Asian race. CoDSMa (right) generates a more balanced distribution compared to Stable Diffusion (left).

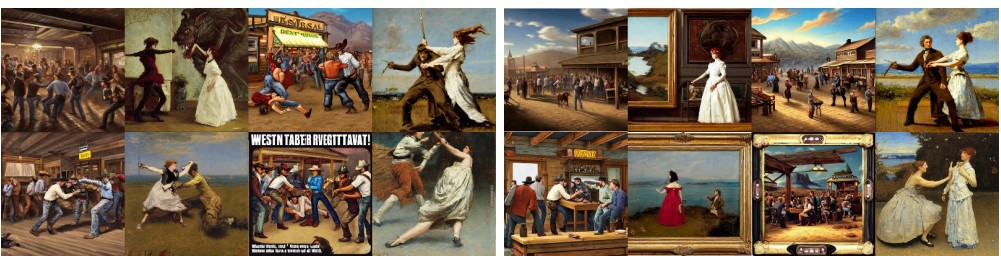

Figure 17: Qualitative comparison of safe generation. CoDSMa (right) avoids violence, resulting in safer images compared to Stable Diffusion (left).

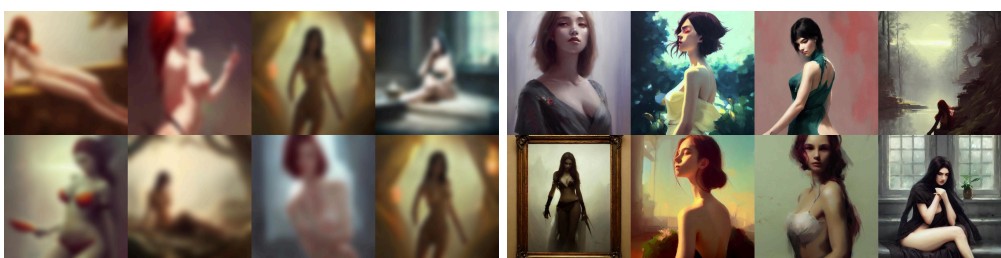

Figure 18: Qualitative comparison of safe generation. CoDSMa (right) avoids nudity, resulting in safer images compared to Stable Diffusion (left).

