# OpenReview forum: "Concept Denoising Score Matching for Responsible Text-to-Image Generation"
_ICLR.cc/2025/Conference — ICLR 2025 Conference Withdrawn Submission_

### Official Review · Reviewer_Zt2Y · 2024-10-27

**Soundness:** 3
**Presentation:** 3
**Contribution:** 2
**Rating:** 5
**Confidence:** 4

**Summary:**

This paper addresses fairness and safety concerns in text-to-image (T2I) diffusion models. Building upon previous works on discovering interpretable concept representations (h-space), it introduces Concept Denoising Score Matching (CoDSMa), which learns responsible concept vectors in the latent h-space of diffusion models. CoDSMa aligns the model’s denoising scores to guide generated content toward fair representations and safe outputs without direct training on biased or harmful examples. Empirical results show that CoDSMa outperforms previous methods in achieving fair and safe T2I generation with minimal impact on image quality.

**Strengths:**

- Fair and safe generation is an important challenge in T2I diffusion.
- Given the interpretability tool (h-space), it is natural to use it to address fairness and safety issues.
- The paper is well-structured and clearly written.

**Weaknesses:**

**Real-time generation?**

Text-to-image (T2I) diffusion often requires real-time image generation. While prior methods utilizing negative prompts or fair/safe guidance can be applied on-the-fly, this work involves finding the c-vector, which may be computationally expensive and challenging for real-time applications.

---
**Limited baselines**

Responsible generation in T2I diffusion has been extensively studied in recent years, as noted in the Background section (see also [1] and related works). However, this paper’s experiments include only a limited subset of these methods for comparison. A broader evaluation of CoDSMa against a wider range of approaches would offer clearer insights into its strengths and limitations, particularly regarding the additional computational cost mentioned above.

[1] Friedrich et al. Fair Diffusion: Instructing Text-to-Image Generation Models on Fairness. arXiv 2023.

---
**Limited novelty**

The core concept of identifying c-vectors is a logical extension of prior work on h-space interpretability. Although the paper introduces some engineering improvements, such as reducing computational cost, the level of technical innovation is not substantial.

**Questions:**

N/A

---

### Official Review · Reviewer_zDW6 · 2024-11-01

**Soundness:** 2
**Presentation:** 1
**Contribution:** 1
**Rating:** 3
**Confidence:** 4

**Summary:**

This paper introduces concept denoising score matching for fair and safe text-to-image generation using diffusion models. Building on the observation that differences in scores based on varying input concepts in the hidden space (h-space) align with the target concept, the authors propose a bias vector, termed the c-vector, in the h-space to adjust for or enhance specific concepts. By combining different c-vectors uniformly, the model achieves a desired distribution over concepts such as gender or race. Experimental results demonstrate the effectiveness of this approach.

**Strengths:**

- The paper addresses fairness and safety in text-to-image generation, an important and timely issue with substantial ethical implications.
- The proposed method is clean and straightforward, yet effective to some extent.
- The motivation is clear and well-articulated.
- The paper is well-organized and includes a thorough literature review.

**Weaknesses:**

Clarity on c-vector: Do the learned c-vectors always add to the h-space, or does this depend on the prompts? For example, with prompts like "a photo of a <profession>," the c-vectors are expected to be helpful for fairness. However, for prompts unrelated to gender, does such a c-vector hurt generation quality? If the c-vectors are not always added to the h-space, how do we know when to add them? This is of significant importance for application, as it would be challenging to decide automatically.

Moreover, in Section 5.3, the composition of fairness concepts is demonstrated, but this raises questions on how to decide which c-vectors to use or drop as more candidates are introduced. This is an important aspect, as the number of relevant c-vectors may grow, and the paper does not address how to select them effectively. This further complicates the previous concern.

Some important evaluations are missing, making the proposed method less convincing. For example, what is the difference between using the learned c-vectors and directly modifying the input prompt with corresponding concepts? Simply appending “a woman” to the prompt might achieve similar effects as using the c-vector for "a woman." A similar approach is proposed by Friedrich et al. 2023, which should be discussed.

The writing in the paper is sometimes confusing, and a thorough revision is needed. Some sections are contradictory or unclear (see a few in my questions).

---

[a] Friedrich, F., Brack, M., Struppek, L., Hintersdorf, D., Schramowski, P., Luccioni, S., & Kersting, K. (2023). Fair diffusion: Instructing text-to-image generation models on fairness. arXiv preprint arXiv:2302.10893.

---

Minor:
- In L254, a word is capitalized after a comma. Further polish of the grammar and spelling is recommended throughout.

**Questions:**

- In Section 4.2, the denoising latent $z\_t$ is generated with the neutral prompt and a pretrained diffusion model. However, according to the text, "reverse diffusion to timestep $t$ with a c-vector and 'a person' prompt yields latent" in Fig. 1, it seems that $z\_t$ is generated using the c-vector. Could you clarify this discrepancy?
- In Line 271, the U-net $\\epsilon\_\\theta$ is described as learnable, yet the target prompt $y\_p$ and $z\_t$ are input to a pretrained U-net $\\epsilon\_\\theta$. The notations here are somewhat confusing. According to Fig. 1, it appears that the only trainable parameters are the c-vectors in the h-space. Could you clarify this point?
- In Eq. (12), the second gradient term seems irrelevant to $c$. Does this imply the gradient is zero?

---

### Official Review · Reviewer_mLuS · 2024-11-03

**Soundness:** 3
**Presentation:** 3
**Contribution:** 2
**Rating:** 6
**Confidence:** 4

**Summary:**

This paper introduces Concept Denoising Score Matching (CoDSMa), an approach for responsible text-to-image (T2I) generation in diffusion models, addressing biases and harmful content. CoDSMa learns responsible concept representations in the bottleneck feature activation space (\textit{h-space}), leveraging prompt alignment to guide score predictions toward desired concepts. Empirical results show that CoDSMa reduces biased and harmful content generation by nearly 50% compared to state-of-the-art methods.

**Strengths:**

1. The research topic in this paper is relevant to the community. It handles both fairness and harmful content generation.
2. The authors did a great summary of related work.
3. The experimental results are promising.

**Weaknesses:**

1. Novelty: The work is an incremental work of [1] SDisc. Similar to SDisc, the goal is to optimise a concept vector to manipulate the generation process. The difference is how to learn the vector. In this work, the author proposes a new loss to learn the vector.

2. Lack of experiments/discussion about efficiency. The author claims the proposed method is an efficient approach.

3. It is great to show some visualisation about the combination of different learned concepts. What about more combination, e.g. three?

4. Since the experimental design is very similar to SDisc. It would be great to show some comparison in the combination of different concepts in Figure 5, and degree of manipulation in Figure 6.

[1] Hang Li, Chengzhi Shen, Philip Torr, Volker Tresp, and Jindong Gu. Self-discovering interpretable diffusion latent directions for responsible text-to-image generation. In CVPR, 2024.

**Questions:**

1. Is Linear transformation the best option when manipulating a single concept or combining different concepts?  More deep discussion about this is expected.

2. What is the intuition Why the proposed one is “significantly” better than SDisc?

3. Does the proposed method also works for new harmful prompts in [2]?

[2] Liu, Tong, et al. "Multimodal Pragmatic Jailbreak on Text-to-image Models." arXiv preprint arXiv:2409.19149 (2024).

---

### Official Review · Reviewer_YaMk · 2024-11-03

**Soundness:** 1
**Presentation:** 3
**Contribution:** 2
**Rating:** 3
**Confidence:** 3

**Summary:**

This paper presents a new approach to advance responsible text-to-image generation, specifically enhancing output diversity and reducing the generation of unsafe images. To this end, the authors build on the work of Li et al. (2024) and aim to decrease computational cost by moving the optimization of latent vectors to influence the sampling process from the image space to latent representations. The approach is evaluated on two benchmarks: Winobias (for fairness) and I2P (for safety).

**Strengths:**

- The novel approach is well introduced, providing all necessary preliminaries on text-to-image diffusion models and the foundation work of Li et al. (2024).

- Next to aiming to decrease computational cost, the approach also seems to increase performance compared to Li et al. (2024).

- In their ethical statement, the authors acknowledge the limitations of their evaluation concerning the considered dimensions of gender and race.

**Weaknesses:**

- The paper lacks a comprehensive limitations discussion. For example, it remains unclear whether the proposed method generalizes to other diffusion model architectures.

- The evaluation strictly follows the protocol from Li et al. (2024), overlooking recent benchmarks, such as the one presented at NeurIPS (https://proceedings.neurips.cc/paper_files/paper/2023/file/b01153e7112b347d8ed54f317840d8af-Paper-Datasets_and_Benchmarks.pdf). Beyond the prompt set (which already appears quite similar to the prompts used in the current study), the clustering-based approach may provide additional insights into the approach’s performance and generalization.

- The paper offers only an incremental extension of Li et al. (2024) to address the computationally expensive process. However, it is not clear to the reader that this is an issue in practice. Especially, although the approach claims to address computational inefficiencies in current methods as the main contribution, the paper does not provide any comparison of computational costs with existing methods. This weakens the claim of reduced computational expenses and the significance of the introduced approach.

**Questions:**

- Does the proposed approach generalize to architectural variations, such as rectified flow-based or diffusion transformer (DiT) models?

- Could the authors provide quantitative comparisons of the computational costs (such as runtime and memory usage) between the proposed method and existing approaches, particularly in comparison to Li et al. (2024)? Additionally, could you elaborate on the practical implications of the computational inefficiencies your method aims to address?

---

### Note · Authors · 2024-11-21

**Comment:**

We appreciate the reviewers for their feedback and suggestions.

**Withdrawal Confirmation:**

I have read and agree with the venue's withdrawal policy on behalf of myself and my co-authors.